# Hawaiian Treeline Ecotones: Implications for Plant Community Conservation under Climate Change

**DOI:** 10.3390/plants13010123

**Published:** 2023-12-31

**Authors:** Alison Ainsworth, Donald R. Drake

**Affiliations:** 1School of Life Sciences, University of Hawai’i at Mānoa, Honolulu, HI 96822, USA; dondrake@hawaii.edu; 2National Park Service, Pacific Island Inventory and Monitoring Network, Hawaii National Park, HI 96718, USA

**Keywords:** vegetation, subalpine, alpine, treeline, Hawai’i Volcanoes National Park, Haleakalā National Park

## Abstract

Species within tropical alpine treeline ecotones are predicted to be especially sensitive to climate variability because this zone represents tree species’ altitudinal limits. Hawaiian volcanoes have distinct treeline ecotones driven by trade wind inversions. The local climate is changing, but little is known about how this influences treeline vegetation. To predict future impacts of climate variability on treelines, we must define the range of variation in treeline ecotone characteristics. Previous studies highlighted an abrupt transition between subalpine grasslands and wet forest on windward Haleakalā, but this site does not represent the diversity of treeline ecotones among volcanoes, lava substrates, and local climatic conditions. To capture this diversity, we used data from 225 plots spanning treelines (1500–2500 m) on Haleakalā and Mauna Loa to characterize ecotonal plant communities. Treeline indicator species differ by moisture and temperature, with common native species important for wet forest, subalpine woodland, and subalpine shrubland. The frequency or abundance of community indicator species may be better predictors of shifting local climates than the presence or absence of tree life forms per se. This study further supports the hypothesis that changes in available moisture, rather than temperature, will dictate the future trajectory of Hawaiian treeline ecotone communities.

## 1. Introduction

Alpine treelines are important ecological features of mountains around the world, and trees within treeline ecotones are predicted to be especially sensitive to climate variability because this zone represents tree species’ altitudinal limits. The alpine treeline ecotone is the transition zone between forest and low-stature subalpine or alpine vegetation, which includes the realized treeline, defined as the line connecting the highest elevation trees of a selected height (typically > 3 m) [1]. Treelines are often conspicuous transition zones [2] and have long captivated biogeographers, with extensive research predominantly focused on temperate alpine regions (reviewed in [1]). Globally, the treeline correlates with average growing season temperatures between 6 and 7 °C for at least 94 days per year [3]. However, in tropical and subtropical regions, especially on islands, treelines occur at warmer isotherms, owing to the small Massenerhebung effect or weak mass elevation effects and the corresponding weaker influence on the vertical temperature gradient compared to large, continental mountain systems [4].

Mountain species on tropical islands may be especially vulnerable to climate change because tropical, high-elevation ecosystems are predicted to experience relatively rapid rates of climate change, thus developing “novel” climates faster than species can adapt to tolerate them [5]. Treeline community resilience on oceanic islands is typically further depressed owing to additional biogeographic and evolutionary processes [6,7,8]. Geologically young, isolated oceanic islands have low species diversity [7,9,10], resulting in depauperate floras with few or no species adapted to high-elevation, cold conditions [4,8]. However, it may be that moisture stress caused by seasonal and/or less-frequent cyclic events (e.g., ENSO-driven drought) is a stronger predictor of treeline position than temperature on tropical islands, such as the Canary Islands, the Dominican Republic, and the Hawaiian Islands [11,12,13,14,15]. The current lack of specific factors that dictate treeline location and dynamics in the tropics severely limits our ability to predict the fate of these endemism hotspots in the face of climate change [16].

While there has been an increase in treeline research since the 1990s—primarily to assess (and largely confirming) the predicted upslope movement of trees with rising global temperatures [17]—few studies have examined tropical systems [18]. When local anthropogenic factors (e.g., wildfire, species invasions) are accounted for, examples of upslope tree migration have been documented in all regions except the tropics [19]. The lack of documented treeline advancement with increasing temperatures supports the hypothesis that temperature is not the only limiting factor for tree growth above the treeline in the tropics; however, the lack of baseline vegetation data in and around tropical treeline ecotones prohibit finer-scale assessments. Multiple authors have highlighted the paucity of data for the tropics and subtropics related to high-elevation species response to climate change and emphasized the need for additional studies on plant distributions to enable predictions about the effects of climate change [20,21]. Where data do exist in south-eastern Australia, the effects of recent anthropogenic wildfires overwhelmed the evidence of climate-driven advancements in the *Eucalyptus pauciflora* treeline [22]. In the Andes of Peru, a 42-year study documented upslope treeline advancement only in protected areas, and movement was <2% of the pace required to remain in equilibrium with the rising mean temperature [23]. Interestingly, the treeline ecotone in this site appeared to act as a hard boundary, in that many tree species had a much higher upslope migration rate (15.5–110%) in the forest below the ecotone than within the ecotone [20,23]. Patterns in tropical treeline ecotone dynamics are complex and depend on local factors, including community composition and structure as well as regional and global climate drivers [19].

To understand the dynamic processes that determine current and future species distributions within tropical treeline ecotones, we must first document the current spatial patterns within these ecosystems [24]. The importance of scale has been increasingly emphasized, with differing tree growth responses to global warming presumably resulting from local and site-specific factors [25]. Although every treeline ecotone is to some extent a unique outcome of combinations of climatic conditions, topography, species composition, and disturbance history [26], treeline ecotones display general spatial patterns, allowing for inference of the underlying processes [27]. For instance, at the local spatial scale, modelling has produced distinct treeline patterns, such as abrupt versus gradual ecotones, only under specific combinations of processes and patterns [27,28]. Mapping species distributions and community composition within ecotones is the critical first step to begin formulating hypotheses about causal ecological processes, and it is necessary to interpret changes in spatial patterns and predict future shifts with ongoing climate change [28].

The objectives of this study were to use a long-term vegetation plot dataset to (1) describe the variability of plant community composition across Hawaiian treeline ecotones and (2) identify environmental variables that are correlated with observed patterns in plant community composition. We hypothesized that moisture is more important than temperature in explaining differences among plant communities within the Hawaiian treeline ecotone regardless of scale. Specifically, we predicted that distinct plant communities exist within the treeline ecotone and that differences among communities are strongly correlated with moisture, such that the treeline occurs at higher elevations (=cooler temperatures) in wetter sites than in drier sites.

## 2. Results

### 2.1. Plant Community Composition

Gamma diversity for the entire study area was 229 species. Three cluster groups were recognized within the Hawaiian treeline ecotone (Figure 1). Diversity measures are reported by cluster group (Table 1), which are hereafter referred to as “subalpine shrubland (SS)”, “subalpine woodland (SW)”, and “wet forest (WF)” based on species composition [29]. The SS included more than triple the number of sample plots (152) of the other groups, yet this imbalance was reflected only in the higher beta diversity measure. Forty-six plots were clustered in the WF, which had the highest species diversity for all other measures, including endemism (67% of WF gamma diversity). Twenty-seven plots were clustered as the SW, all of which were located on Mauna Loa, and non-native species dominated (57%) gamma diversity in this community. The multi-response permutation procedure (MRPP) used to test the null hypothesis of no difference between cluster groups [30] revealed significant differences between these three cluster groups (chance-corrected within-group agreement *A* = 0.157 *p* < 0.001), suggesting that they are distinct species assemblages.

The nonmetric multidimensional scaling (NMS) ordination was used to delineate patterns between sites, clusters, species, and environmental variables [31,32]. The NMS showed remarkably little overlap of the three cluster groups in the species space, with all three separating on the first and second axes (Figure 2). The 3D solution represented 80% of the variance in the dataset (axis 1 r^2^ = 0.33, axis 2 r^2^ = 0.26, axis 3 r^2^ = 0.21), with low final stress (13.98) and instability (<0.001) after 102 iterations.

Each community (=cluster group) was characterized by different indicator species (ISA) based on species’ relative abundance within a particular cluster group and frequency or faithfulness of occurrence of a species in a particular group (Table 2) [30,33]. WF had the greatest within-group agreement among sample plots and contained the greatest number of indicator species (26, including 9 trees, 9 ferns, 4 shrubs, 1 herb, 2 grasses, and 1 vine) representing 18 families. WF indicator species were all endemic (21 spp.) or indigenous (5 spp.), and most were restricted to the wet forest (Figure 3a,b), including trees (e.g., *Cheirodendron trigynum* ssp. *trigynum*), tree ferns (e.g., *Cibotium glaucum*), and many ferns. The SS community had the lowest within-group agreement among sample plots, as demonstrated by high beta diversity. In total, 14 indicator species were identified, representing 9 families (5 shrubs, 2 ferns, 1 herb, 5 grasses, and 1 rush), including endemic, indigenous, and non-native species. Many strong indicators of the SS community were also found in the other communities (Figure 3c–e), but in greater abundance and at a higher frequency in the SS community (e.g., *Vaccinium reticulatum, Deschampsia nubigena, Leptecophylla tameiameiae*). The SW community had intermediate within-group agreement and 15 indicator species, of which 11 were non-native. SW indicator species represented 11 families (2 trees, 1 shrub, 10 herbs, and 2 grasses). The strongest indicator was an abundant, non-native grass, *Ehrharta stipoides* (Figure 3f), but two native shrub/tree species were also strong indicators: indigenous *Myoporum sandwicense* and endemic *Sophora chrysophylla* (Figure 3g,h). A comparison of Jaccard habitat specialization index values per indicator species (Table 2) among the three community types revealed a significantly higher habitat generalist value for the SW community (*F* = 7.27 *p* = 0.002, Figure 4), likely attributable to the high number of non-native indicator plant species [34].

**Figure 2 plants-13-00123-f002:**
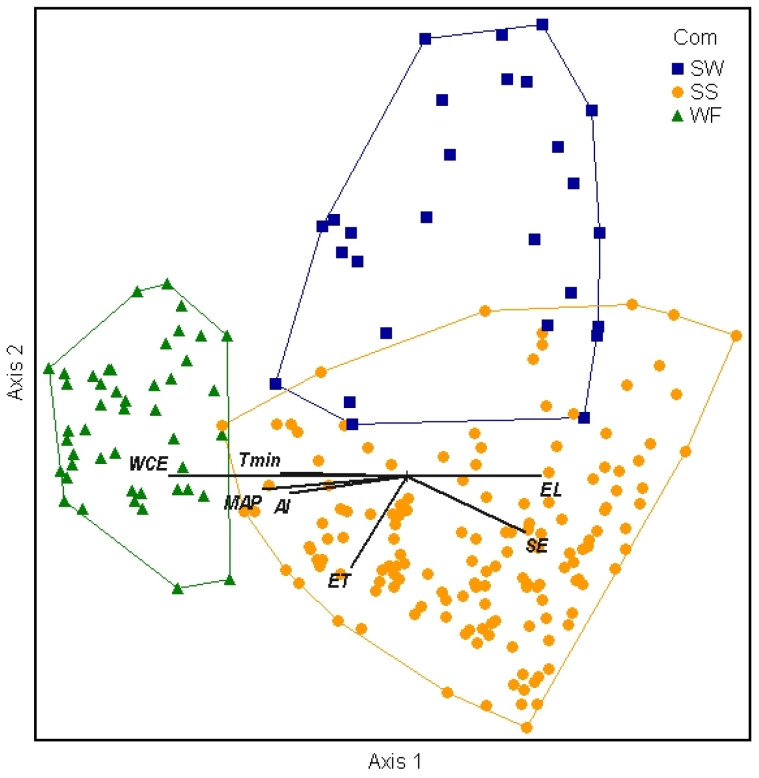
Diagram of NMS ordination analysis along axes 1 and 2 showing the positions of separated cluster groups (SW = subalpine woodland, SS = subalpine shrubland, WF = wet forest). Environmental variables with significant correlations are displayed as joint plot overlays (see Table 3). After rotation to maximize the correlation of elevation (EL) with axis 1, soil evaporation (SE) was also positively correlated with axis 1. Wet canopy evaporation (WCE), mean annual precipitation (MAP), minimum temperature (Tmin), and aridity index (AI) were negatively correlated with axis 1. Evapotranspiration (ET) was the only strong correlate with axis 2.

### 2.2. Plant Community–Environment Correlations

Strong correlations (*r* > 0.40) of environmental variables with axis 1 included both positive (elevation and soil evaporation) and negative (wet canopy evaporation, mean annual precipitation, minimum temperature, and the aridity index) relationships (Table 3, Figure 2). Evapotranspiration was the only environmental variable notably correlated (*r* = −0.42) with axis 2. Categorical environmental and site-characteristic variables were generally not as important in explaining the observed plant community patterns as the continuous variables at this scale (Figure 5a–e). For each categorical variable, plots in the ordination space were coded by the variable and enclosed by a loop or arc to assess whether the categorical variable is patterned on the ordination. The tree (>3 m) category grouping was not consistent with the cluster community groups due to the presence of large *Metrosideros* trees in many (44) subalpine shrubland plots on Mauna Loa. Substrate age and lithology categories do not explain the community clustering observed at this scale, as indicated by the lack of separation of groups. Similarly, sample plot size and volcano do not correlate with plant community patterns, with the one exception that the SW community was documented only on Mauna Loa.

The WF had higher moisture variables (wet canopy evaporation, mean annual precipitation, aridity index) and warmer temperatures (mean annual temperature, minimum temperature). Conversely, the subalpine communities were often at higher elevations and cooler temperatures, with greater soil evaporation. Two of the three strongest correlates, wet canopy evaporation and elevation, did not differ by volcano. Mean annual precipitation, mean annual temperature, minimum temperature, aridity index, soil evaporation, and evapotranspiration all differed by community and volcano.

### 2.3. Site-Specific Patterns

#### 2.3.1. Haleakalā

In total, 156 plant species were documented across the 62 sample plots on Haleakalā. Only WF and SS communities were found within the sample frame. NMS analysis of Haleakalā alone produced a robust 2D solution representing 81% of the variance in the dataset (axis 1 *r^2^* = 0.57, axis 2 *r^2^* = 0.24), with low final stress (12.91) and instability (<0.001) after 52 iterations. Environmental variables differed dramatically between these two communities (Figure 6a). Grouping plots by canopy trees correlated closely with the cluster groups (Figure 6b), unlike the full regional dataset. On the windward side (northeast aspect), wet forest community plots reach elevations above 2078 m before transitioning into the subalpine shrubland, whereas on the leeward side (southeast aspect), the subalpine community persists below 1531 m (Figure 7a). This site-specific spatial pattern corresponds with modelled mean annual precipitation (Figure 7b), except for three unusually wet SS plots. Two are within rare bog sites and the other is in a valley depression where forest growth is likely limited by cold temperatures (Figure 7c). The strongest correlates dividing these communities on Haleakalā were higher wet canopy evaporation (*F* = 62.79 *p* < 0.001, Figure 8a) and higher mean annual precipitation (*F* = 43.9 *p* < 0.001, Figure 9a) in the WF community compared to the SS community. On Haleakalā, the threshold where WF transitions to SS is between 200 and 400 mm for wet canopy evaporation (decreasing with elevation (Figure 8b)) and 4000 mm mean annual precipitation (independent of elevation (Figure 9b)).

#### 2.3.2. Mauna Loa

In total, 149 plant species were documented across the 163 sample plots on Mauna Loa. WF, SS, and SW communities were found within the sample frame. NMS analysis of Mauna Loa alone produced a robust 3D solution representing 83% of the variance in the dataset (axis 1 *r^2^* = 0.34, axis 2 *r^2^* = 0.35, axis 3 *r^2^* = 0.14), with low final stress (12.92) and instability (<0.001) after 82 iterations. Environmental variables differ among communities, similarly to the full regional dataset analysis (Figure 10a). Tree cover is less informative than community composition for grouping plots (Figure 10b), as in the full regional analysis. Plots in the SW community were found only on Mauna Loa, and they were adjacent to SS in just three distinct sites on the volcano (Figure 11a). The treeline ecotone between WF and SS corresponds to mean annual precipitation, as on Haleakalā (Figure 11b). The SW was found only in dry sites, and the transition to SS may be correlated with local site temperature (Figure 11c). Mean annual precipitation was the strongest correlate with axis 1, with high values in the WF and low values in the SW (*F* = 58.31 *p* < 0.001, Figure 12a). Similarly, the aridity index differed among communities (*F* = 42.36 *p* < 0.001, Figure 13a). The transition from WF to SS on Mauna Loa occurs at approximately 2000 mm mean annual precipitation (Figure 12b), with an aridity index of 1.0 (Figure 13b), independent of elevation. Identifying a threshold trigger for the transition between SS and SW on Mauna Loa is less clear with these data, as site differences among the three SW locations appear stronger than common patterns among sites. All three SW sites are at different elevations, with different aspects and precipitation values (circled in Figure 11).

## 3. Discussion

Three distinct plant communities—wet forest, subalpine shrubland, and subalpine woodland—were identified based on plant species composition and abundance within the treeline ecotone of Haleakalā and Mauna Loa. These communities differ dramatically in composition and structure, with significantly greater species diversity and structural complexity in the wet forest than the drier subalpine communities. Hawaiian treelines are relatively diffuse (sensu [17]), and we suggest that the presence of community indicator species and/or their frequency or abundance are more sensitive to shifting climatic conditions in this region than the treeline or the presence or absence of tree life forms per se. Moisture explains the described patterns in plant community composition better than temperature, with wet canopy evaporation, mean annual precipitation, and aridity index values differentiating between wet forest and subalpine communities. Based on this relationship, changes in available moisture, rather than temperature, will likely dictate the future trajectory of Hawaiian treeline ecotone communities.

### 3.1. Treeline Ecotone Composition and Structure

#### 3.1.1. Wet Forest

The WF was the most homogeneous community within the treeline ecotone despite having the greatest species diversity. The WF was dominated by native species with very high endemism (67%) and low non-native species richness (13%). The diversity values are useful comparatively; however, because these numbers were generated from variably sized sample plots (i.e., plot size influences *α* richness) and sample sizes per community (i.e., the sample size or total area sampled influences *β* and *γ* diversity), caution should be used if applying diversity values outside of this study. Importantly, plot size does not correlate to patterns in community separation (Figure 5d), and the SS community with the greatest sample size did not have the greatest diversity values. 

Most WF indicator species were found exclusively in the WF community. Due to this connection, observing one of these species establishing in subalpine communities may suggest early community movement associated with climate change. *Cheirodendron trigynum* ssp. *trigynum* was the strongest WF indicator of the treeline ecotone because this endemic tree was found in all 46 sample plots in high abundance (>6% absolute canopy cover). *Cheirodendron* is a bird-dispersed, common habitat generalist of the Hawaiian mesic and wet forests [34,35]. Similarly, the presence of *Cibotium glaucum*, an endemic tree fern within drier (or formerly drier) sites, would support increasing microclimatic moisture conditions. Although *Cibotium* was found in only half of the WF plots, it requires mesic to wet conditions, and therefore its presence indicates these conditions. Many of the other WF indicators are mesic–wet, requiring ferns and some shrubs. Most Hawaiian ferns within the treeline ecotone require mesic to wet conditions, with some drier subalpine exceptions (e.g., *Pellaea ternifolia*) [36]. The WF does include as indicators some species that are found in all communities, such as the Hawaiian endemic canopy dominant *Metrosideros*. The dominant tree was found in all three treeline ecotone communities, which was expected considering this species’ incredible habitat range [35] and ranking as the strongest native habitat generalist [34]. This species is designated as a WF indicator because of its extremely high abundance within the WF community (>50% canopy cover). The presence of *Metrosideros* alone does not indicate increased moisture; however, the presence of a very high canopy cover may. The WF has the highest structural complexity with consistent canopy tree layers; at times, emergent canopies; and, in many sites, tertiary tree fern canopy layers [37]. Epiphytes and hemiepiphytes are important components of WF communities, with most fern species and even *Cheirodendron* (among other trees) establishing in the crowns of *Metrosideros* [36,37]. Epiphytes have been highlighted as some of the best early indicators of changing climatic conditions in the tropics and subtropics [38,39,40]. To increase early detection of changing climatic or microclimatic conditions within the treeline ecotone, more detailed analyses of epiphyte composition should be included.

#### 3.1.2. Subalpine Shrubland

The SS was the most widespread and heterogeneous community within the treeline ecotone, likely due to it having the largest sample size. As a result of three times the number of sample plots, the gamma diversity was nearly as high in this community as the WF, yet the alpha richness remained low. Despite the SS community’s higher percentage of non-native species (45%), this community still had high species endemism (39%). Species invasions in these high-elevation sites are ongoing [41]; currently, cover values remain low [42], but because many non-natives can increase exponentially and be disruptive to native communities, early detection and rapid response are critical management actions for these high-elevation communities [43]. 

The SS indicator species were found in other communities of the treeline ecotone, but they were identified as indicators of this community due to their high frequency and abundance. The lack of tree life forms is not an adequate descriptor for this community, because many plots in a portion of Mauna Loa contained tall *Metrosideros* trees. Unlike the situation with WF indicator species, the presence of SS indicator species in other communities does not suggest shifting climatic or microclimatic conditions. 

The strongest SS indicator is the endemic shrub *Vaccinium reticulatum* owing to its consistently high abundance within plots (>5% absolute cover). *Vaccinium* is extremely cold-tolerant [44] and, overall, it is a strong habitat generalist (top 30%) despite loyalty to dry environments [34]. Similarly, the indigenous shrub *Leptecophylla tameiameiae* is a strong habitat generalist (top 14%) and an indicator of the SS community because it has higher frequency and cover within this community than in others of the treeline ecotone [34]. *Leptecophylla* is an extremely wide-ranging, bird-dispersed shrub found from the coast to above the treeline [35]. Subtle differences in the abundance of *Leptecophlla* have been used to distinguish park-wide vegetation maps [45,46]. The endemic graminoids *Deschampsia nubigena* and *Morelotia gahniiformis* are also strong SS indicators, but unlike the shrubs, these are closer to the habitat specialist end of the spectrum [34]. *Deschampsia* is described as present in mesic–wet environments, yet it was abundant within this community, whereas *Morelotia* is described as limited to dry environments [35]. 

The SS community has the least structural complexity within the treeline ecotone. Most sites lack a consistent canopy tree layer, except portions of Mauna Loa with tall *Metrosideros* trees. These sites limit our ability to use *Metrosideros* tree canopy as a sensitive indicator of shifting climates. Although the understory composition within these plots is similar to that of the rest of the SS plots, the presence of tall canopy trees presents more questions for future studies. It is likely that these tall individuals are linked to lava flow age, composition, lithology, and/or other edaphic site factors that were not evident at the scale of this study. Additional site-specific analyses may be important to understand underlying processes explaining these differences and, therefore, to strengthen our ability to predict future community composition under shifting climatic conditions.

#### 3.1.3. Subalpine Woodland

The SW community was recorded only on Mauna Loa in three geographically disparate sites (Figure 11). This community had the least number of sample plots and, correspondingly, the the lowest gamma diversity; however, other measures of diversity were higher than the SS community, possibly due to increased structural complexity. The SW community had patchy structural complexity, with dry, open stands and, often, patches of rough aa substrate. The SW community was the most anthropogenically disturbed community (e.g., feral sheep, mouflon, goats), with more than half of the species recorded as non-native, yet this community still contained high (30%) endemism. 

The strongest SW indicator is the invasive grass *Ehrharta stipoides,* which was present in high abundance (>35% absolute canopy cover) in every SW plot. This grass was ranked as a very high habitat generalist (top 8%; [34]), and it was the only non-native species frequently encountered along monitoring transects in both subalpine and wet forest sampling frames of Hawai’i Volcanoes National Park [42]. This invasive grass thrives in multiple sites in part due to higher shade tolerance compared to other invasive grasses [47]. Hawaiian habitat descriptions initially described *Ehrharta* as invading mid-elevation mesic/wet forest openings, which was unusual for this species [48]. Additional sightings now include higher elevations in drier sites on Haleakalā and Mauna Loa [42,49] and invasions onto dry sites of additional islands (e.g., Kaho’olawe, Kaua’i) [50]. All but four of the other SW indicators were non-native grasses or herbs that had high habitat generalist values, which explains this community’s high generalist affiliation (Figure 4) [34].

Indigenous *Myoporum sandwicense* and endemic SW indicators *Sophora chrysophylla* and *Acacia koa* are strong habitat generalists (top 30%) found from dry to wet sites [34,35]. These native shrub/trees in the SW community have all been negatively influenced by past land use and the history/presence of non-native ungulates (e.g., mouflon sheep, goats, cattle). Previous Hawaiian treeline descriptions have identified a treeline zone of *Myoporum* and *Sophora* above the *Metrosideros* zone on Haleakalā and Mauna Kea. The absence of this feature on Mauna Loa was hypothesized to be related to the younger substrate of Mauna Loa and the lack of pyroclastic deposits [4]. Interestingly, we found this assemblage of species only on Mauna Loa. The drier portions of Haleakalā that likely contain these assemblages were either outside of this sample frame or had been heavily disturbed by non-native ungulates (e.g., Nu’u Unit on the south flank) [51]. Thus, we cannot definitively characterize Hawaiian treeline limits.

Community clusters derived from species composition and abundance are better indicators of future climatic changes than trees alone for the full dataset (Figure 5a) because of the complication of tall *Metrosideros* trees on Mauna Loa in the SS community. Additionally, the volcano (Figure 5e) fails to represent the compositional and structural differences between the SS and SW communities on Mauna Loa. Indicator species and compositional descriptions are useful, but to make predictions and assess potential management actions with shifting climatic projections within the treeline ecotone, we must identify critical environmental thresholds between communities.

### 3.2. Environmental Drivers

The distributions of the three different plant communities within the Hawaiian treeline ecotone on Haleakalā and Mauna Loa are best explained by moisture status. Previous studies demonstrated how drought associated with strong El Niño events limit the wet forest’s upper edge near the TWI on windward Haleakalā [11,52]. We hypothesized that moisture would also limit tree distribution at the upper edge of the treeline ecotone across a range of environmental conditions, including windward and leeward aspects on Haleakalā and Mauna Loa. These findings support this general hypothesis at the regional scale. However, in order to identify threshold moisture values distinguishing among communities, volcano-specific analyses were required due to the dramatic differences in moisture between islands.

Wet canopy evaporation (WCE) was the strongest environmental correlate with axis 1 and divided the wet forest from the subalpine communities. This modelled moisture variable incorporates the important fog/mist conditions common in these wet forests near the TWI with the standard mean annual precipitation (MAP) values. On Haleakalā, WCE decreases with elevation, with the WF to SS transition between 200 and 400 mm per year (Figure 8). On Mauna Loa, the threshold WCE range where WF transitions to SS is between 200 and 300 mm; however, no common threshold was identifiable between SW and SS communities owing to significant site differences within the SW community. 

As expected, MAP is also important in defining community transitions, and it tends to be a more accessible variable to base management decisions on than WCE. Overall, plots on Haleakalā were much wetter than those on Mauna Loa. Haleakalā includes the true windward zone, whereas Mauna Loa is partially in the rain shadow of Mauna Kea, with some leeward plots also in the rain shadow of Mauna Loa. Additionally, more of the plots on Mauna Loa had south- and west-facing aspects. The MAP threshold between WF and SS for Haleakalā was 4000 mm, independent of elevation (Figure 9), whereas the threshold dividing WF and SS on Mauna Loa was half that (2000 mm) and decreased with elevation (Figure 12). Similarly to WCE, MAP was not sensitive enough to distinguish between the three disparate sites of SW and SS on Mauna Loa.

The third important moisture indicator that explained variance in the distribution of plant communities in the treeline ecotone was the aridity index (AI). Like WCE, this index is also derived from MAP, but by dividing by potential evapotranspiration, this index highlights sensitivity to dryness. Based on AI, both SS and WF communities are far more sensitive to drying on Haleakalā than on Mauna Loa. This supports the concept that drought impacts from climate change may be more severe for “drought-naïve” populations or those well within their ecological moisture limits. Theoretically, these populations may be more vulnerable due to some combination of physiological (e.g., lack of within-species drought response trait variation) and ecosystem-level (e.g., tree density or leaf area) limitations [53]. Alternatively, populations within the treeline ecotone on Mauna Loa may be more at risk of drought-induced mortality simply because individuals may already be maximizing compensating mechanisms (e.g., physiological adaptation and plasticity) and are therefore existing at their physiological limits, and further drying would cause populations to cross beyond the species’ fundamental niche [53]. This is a critical need for future studies to address in order to triage native Hawaiian high elevation conservation areas in the face of ongoing drying trends [54].

Clearly, moisture availability is a critical plant community driver within the tropical treeline ecotone. Interestingly, our prediction of higher elevation treelines in wetter sites did not hold across the full regional dataset. For example, sites on Haleakalā were much wetter than sites on Mauna Loa, yet the treeline, or where WF and/or SW transition to SS, were also higher on Mauna Loa. It is important to note that Mauna Loa is >1100 m taller than Haleakalā, and montane transition zones are lower on shorter mountains [7]. This highlights the importance of scale. In the previous windward Haleakalā treeline study, within the two-kilometer study area, the treeline was at a higher elevation along the wetter portion [11]. Similarly, across the larger Haleakalā study area, we found support for this pattern when overlaying sample plots on the MAP map (Figure 7). It is evident that WF plots are above 2000 m before transitioning to SS in the northern, windward portion of the study area, whereas SS plots extend down to 1500 m in the leeward, southern portion of the study area. These suggestive patterns require additional site-specific data, particularly on Mauna Loa, to determine exactly how and where moisture limits the treeline between the SW and SS communities. 

Temperature and its associated variables are also important for explaining patterns within the treeline ecotone, albeit to a lesser extent than moisture variables. Elevation serves as a surrogate for temperature on steep volcanic islands like these with wet adiabatic lapse rates of 6 °C per 1000 m in elevation. Elevation and minimum monthly temperature are strong correlates with axis 1, thus dividing WF and subalpine communities. Similarly to extreme drought events limiting tree growth within the treeline ecotone, extreme temperature lows, as captured by the monthly minimums, may exceed tree species’ physiological cold tolerance, as has been previously described for *Metrosideros* and *Acacia* on Hawai’i Island [55,56,57,58]. Interestingly, minimum temperatures on Hawaiian volcanoes are close to the global treeline annual growing season averages of 6–7 °C [3]. The mean annual temperatures of sample plots were much higher than global treeline averages, as expected for oceanic islands [14]. Between volcanoes, Haleakalā had colder SS and warmer WF minimum and mean annual temperatures than Mauna Loa. These differences highlight the potential importance of scale for temperature and also how microclimatic temperature conditions may differ because of microtopography (e.g., Figure 7c on Haleakalā and possibly the SS/SW boundaries in Figure 11c on Mauna Loa). Considering that warming—particularly at high elevations—is already occurring on the Hawaiian Islands [59,60], temperature may become even less important of an influence on treeline in the future. Even in temperate–treeline models, temperature is expected to cross a threshold this century, at which point other abiotic and biotic controls will limit treelines as opposed to temperature [19]. 

Many other abiotic and biotic factors (e.g., pollinators, seed dispersal, propagule pressure, competition, facilitation, invasive species, herbivory) affect plant community composition and distribution within the Hawaiian treeline ecotone. Soil age and lithology are clearly important state factors influencing plant communities, but at the scale of this study, their role was not evident. With additional site-specific modeling, such as that needed to untangle the environmental differences between SS and SW on Mauna Loa, it is likely that these and other edaphic factors will become relevant. Soil development takes time, and with rapidly shifting climatic conditions at high elevations, there are many uncertainties surrounding if and to what extent soil conditions may limit plant communities’ capacities to migrate.

## 4. Materials and Methods

This study was conducted on the Hawaiian Islands, USA (19–22° N, 155–160° W; Figure 14). The Hawaiian Islands have three shield volcanoes tall enough to form distinct treeline ecotones: Haleakalā (3055 m) on Maui Island and Mauna Kea (4207 m) and Mauna Loa (4169 m) on Hawai’i Island. While these mountains share many abiotic and biotic features (e.g., volcanic substrate, species), plant community patterns and ecological processes differ on the volcano and local site scales. Previous studies have highlighted the Hawaiian Islands as the classic example of extreme isolation resulting in a depauperate flora lacking native tree species adapted to high-elevation, cold temperatures and thereby generating a “false treeline” [61]. The Hawaiian native species treeline is lower in elevation and warmer than the physiological limit for tree growth, as evidenced by the presence of invasive *Pinus* and *Eucalyptus* tree species at elevations well above the native treeline in the subalpine shrublands [51,62,63]. Additional studies hypothesized that tree growth above the treeline ecotone on windward Maui is limited by seasonal drought associated with the strong trade wind inversion (TWI) rather than by temperature above the treeline [11,13]. In contrast, on Hawai’i Island, studies have shown the physiological effects of freezing events on dominant Hawaiian tree species *Metrosideros polymorpha* and *Acacia koa* and proposed temperature as limiting tree growth [55,56,57,58]. Others have suggested that the lack of soil development on Mauna Loa, with its younger lava fields and sparse pyroclastic deposits, explains the lack of a distinct upslope belt of *Sophora chrysophylla* above the *Metrosideros* treeline [4]. Interpreting the community and ecosystem effects of the rapidly changing climate on treeline communities within the Hawaiian Islands [59,64,65,66] is challenging without a good understanding of the processes driving species and community distribution within the treeline ecotone.

Sampling occurred in National Parks on the islands of Hawai’i and Maui. Sites range from 1500 m to 2500 m a.s.l. Mean annual precipitation ranges from 695 to 10,208 mm and mean annual temperature ranges from 8.4 to 13.6 °C [67,68]. Climatic variation within the treeline ecotone is driven by north-easterly trade winds, which bring moisture to the slopes just below the trade wind inversion layer (TWI). The TWI is a temperature inversion that persists through most of the year (>80%) and caps cloud vertical development (base heights 2076–2255 m), resulting in sharp decreases in humidity above it [69]. On windward (north and/or east) sides of the mountains, the TWI corresponds to the upper limit of the wet forest [70,71]. The Hawaiian Islands are volcanic in origin, and sampled substrates range from less than 400 years old on Hawai’i to over one million years old on Maui. Sample sites are located on andisol and aridisol soil orders [72], including 30 different soil series [73]. Sample sites are on slopes angled up to 35 degrees facing multiple aspects (Haleakalā—N, NE, E, SE; Mauna Loa—E, SE, S, SW, W) and include five lithology categories: aa, pahoehoe, aa + pahoehoe, tephra fallout + ash, and sand + cobbles [73].

Vegetation near the treeline on Haleakalā and Mauna Loa includes montane and subalpine community types, based on elevation, that range from dry *Deschampsia* grasslands and *Leptecophylla* shrublands to wet *Metrosideros* forests [29]. Recently, both National Parks were mapped to the plant association level, and more than 30 distinct plant associations were defined near the treeline [45,46]. Although all sites are currently protected natural areas, non-native ungulates and plants have altered these communities to varying extents.

We analyzed two vegetation datasets totaling 225 plots and 229 plant species. Data were collected by the National Park Service Pacific Island Inventory and Monitoring Program within Hawai’i Volcanoes and Haleakalā national parks on Hawai’i and Maui between 2005 and 2012 (archived by Pacific Island Inventory and Monitoring, Hawaii National Park, HI, USA). All plots included in this analysis were near the treeline (<2 km from the 2000 m contour; between 1504 and 2498 m elevation) on Haleakalā and Mauna Loa. One dataset contains 68 circular (400 m^2^) and 100 rectangular (700 m^2^) plots installed in relatively homogenous vegetated areas (>5% cover) for vascular plant inventories and classification of plant communities [45,46]. Plant species’ presence and abundance were recorded as foliar absolute cover relevé estimates [74]. The second dataset contains 57 rectangular plots (1000 m^2^) installed in wet forests, subalpine shrublands, and subalpine woodlands to assess the status and detect long-term trends within upland plant communities [75]. Plant species presence and abundance were recorded using modified Whittaker nested plots [75]. Species’ presence and tree density data were converted to absolute cover to enable comparisons with the vascular plant inventory dataset (Appendix A and see [76,77]). Plots were also identified by the presence of tree life forms greater than three meters in height in three categories (treeless, <10% tree cover, ≥10% tree cover) to enable comparison with global treeline assessments. Plant nomenclature, biogeographic origin, and life form were standardized [35,36,50].

Mean annual precipitation, mean annual temperature, minimum and maximum temperature, Penman–Montieth-modeled potential evapotranspiration, transpiration, wet canopy evaporation, soil evaporation, and evapotranspiration were calculated for each site [67,68]. An aridity index (AI) was calculated as the mean annual precipitation/potential evapotranspiration to further assess plant community composition sensitivity to dryness following methods detailed in a recent study of *Metrosideros* leaf trichomes in Hawai’i [78]. The lower the index, the more arid the conditions: AI < 0.03, hyper-arid zone; 0.03 < AI < 0.20, arid zone; 0.20 < AI < 0.50, semi-arid zone; 0.50 < AI < 0.75, dry, sub-humid zone; AI > 0.75, humid zone [79]. Substrate age and lithology were assigned categorically for each site [73]. Environmental data for each sample site are available in the Appendix A. 

Multivariate statistical analysis of plant community data is based on cover values for all species [30]. Through cluster analysis using the Sorensen (Bray–Curtis) distance measure with a group average linkage method, we divided the 225 sites into three clusters. A multi-response permutation procedure (MRPP) was used to test the null hypothesis of no difference between cluster groups. Through indicator species analysis (ISA), we examined the relationships of individual species to cluster groups [33]. Habitat specialization values derived from Jaccard diversity indices [34] were assigned to each indicator species to compare cluster groups’ sensitivity to climate change. Habitat specialization values range from 0.5 (specialist) to 1.0 (generalist). Nonmetric multidimensional scaling (NMS) ordination was used to delineate patterns between sites, clusters, species, and environmental variables [31,32]. Analyses were conducted for the full regional dataset and each volcano independently to compare differences by scale. Multivariate analyses were carried out using PC-ORD [30].

Analysis of variance (ANOVA) and Tukey’s multiple comparison tests were used to further examine relationships identified in the multivariate analysis between plant community composition cluster groups and environmental conditions. All frequentist statistical analyses were carried out using R Statistical Software (v.3.6.3) [80].

## 5. Conclusions

We described three distinct plant communities—wet forest, subalpine shrubland, and subalpine woodland—within the Hawaiian treeline ecotone. We suggest that the presence of community indicator species and/or their frequency and abundance are more sensitive to shifting climatic conditions in this region than the treeline or the presence or absence of tree life forms per se. Moisture variables best explain the described patterns in plant community composition, with coarse threshold wet canopy evaporation, mean annual precipitation, and aridity index values distinguishing between wet forest and subalpine communities. This study further supports previous smaller-scale studies emphasizing the greater influence of moisture compared to temperature within the Hawaiian treeline ecotone and the need to continue improving models that project moisture conditions under different climate change scenarios. It is also important to continue investigating the spatial patterns and underlying ecological processes that divide the subalpine shrubland and woodland community types on Mauna Loa and, likely, Mauna Kea as well to understand the absence of this community on Haleakalā to support future conservation efforts in these disappearing, high-elevation communities.

## Figures and Tables

**Figure 1 plants-13-00123-f001:**
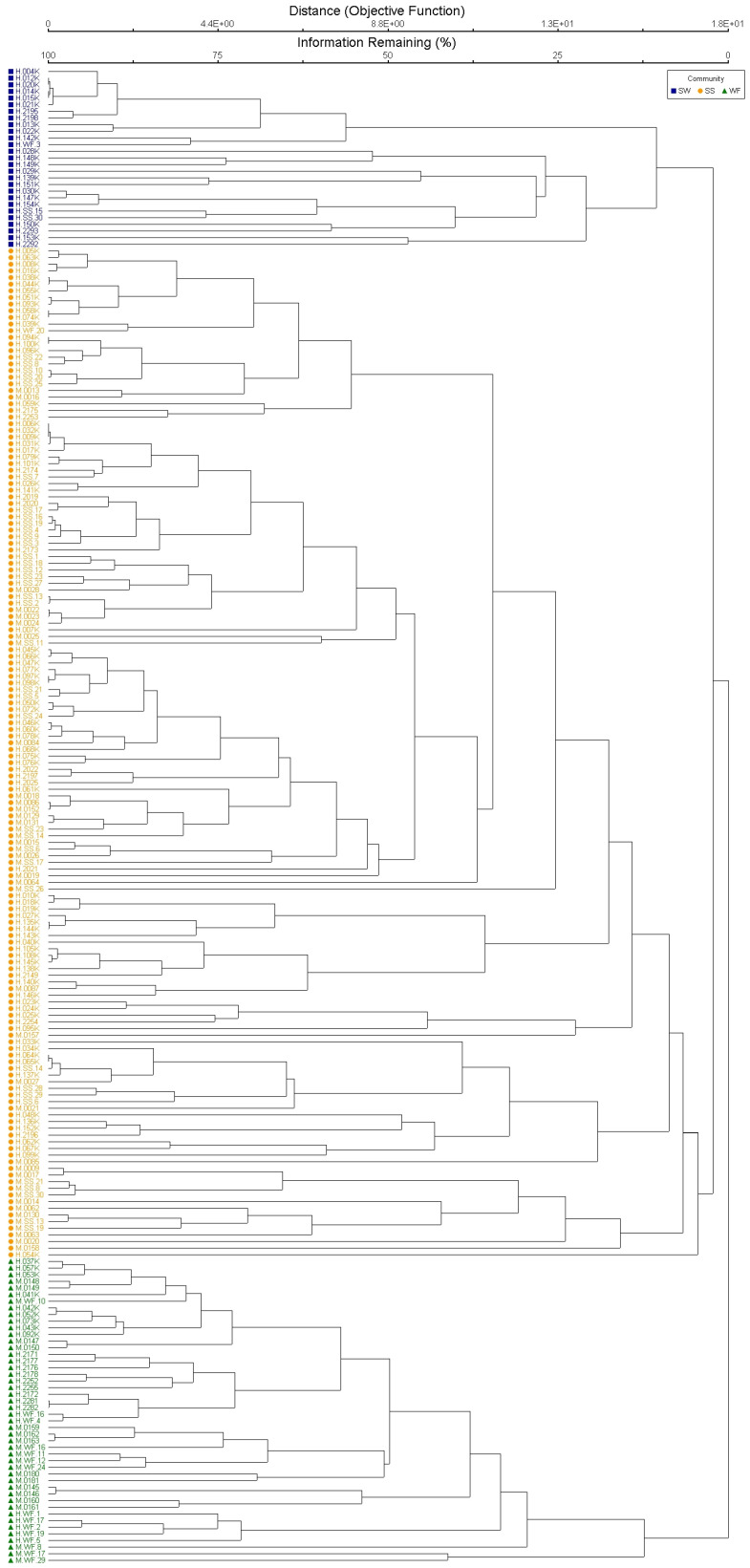
Bray–Curtis classification diagram of the 225 sample plots on Mauna Loa (M) and Haleakalā (H) volcanoes, showing the three separated cluster groups: subalpine woodland (blue squares), subalpine shrubland (yellow circles), and wet forest (green triangles).

**Figure 3 plants-13-00123-f003:**
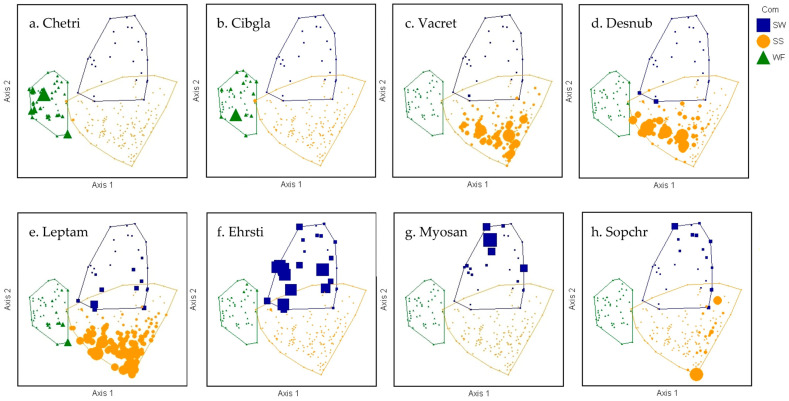
Diagrams of NMS ordination analysis along axes 1 and 2 showing the positions of separated cluster groups (SW = subalpine woodland, SS = subalpine shrubland, WF = wet forest) with indicator species overlays. Within ordination diagrams, larger plot symbols represent higher species abundance. Wet forest indicator species, *Cheirodendron trigynum* (**a**) and *Cibotium glaucum* (**b**), were exclusively found in plots within the WF cluster. Subalpine shrubland indicator species, *Vaccinium reticulatum* (**c**), *Deschampsia nubigena* (**d**), and *Leptecophylla tameiameiae* (**e**), were most frequent and abundant in the SS community. Similarly, non-native *Ehrharta stipoides* (**f**) was found in all types but is far more abundant in the subalpine woodland. *Myoporum sandwicense* (**g**) was only found in the SW community, whereas *Sophora chrysophylla* (**h**) was abundant in two SS plots, but it indicates the SW cluster due to high frequency among SW plots.

**Figure 4 plants-13-00123-f004:**
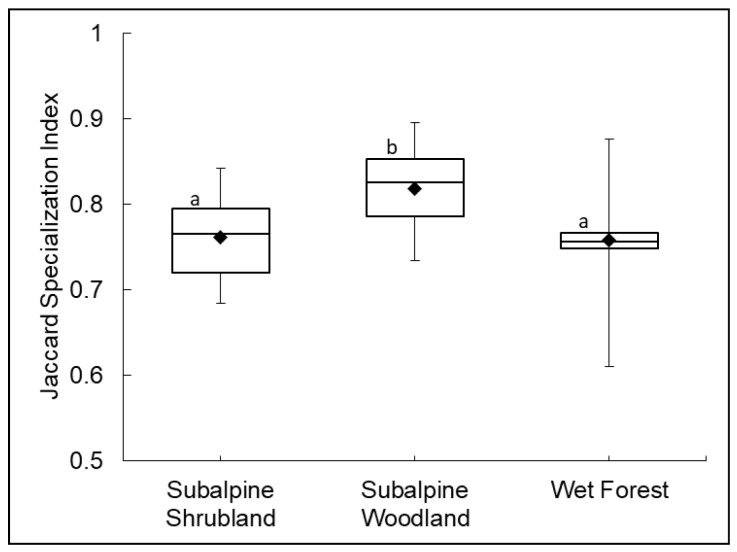
The average Jaccard habitat specialization index of indicator species (Table 2) differs by community. Indicator species in the subalpine woodland (N = 15) are significantly more likely to be habitat generalists than those species that indicate the subalpine shrubland (N = 14) and wet forest (N = 26) communities. Values range from specialist (0.5) to generalist (1.0) based on occupancy and species co-occurrence data [34]. Communities that share the same letter are not significantly different.

**Figure 5 plants-13-00123-f005:**
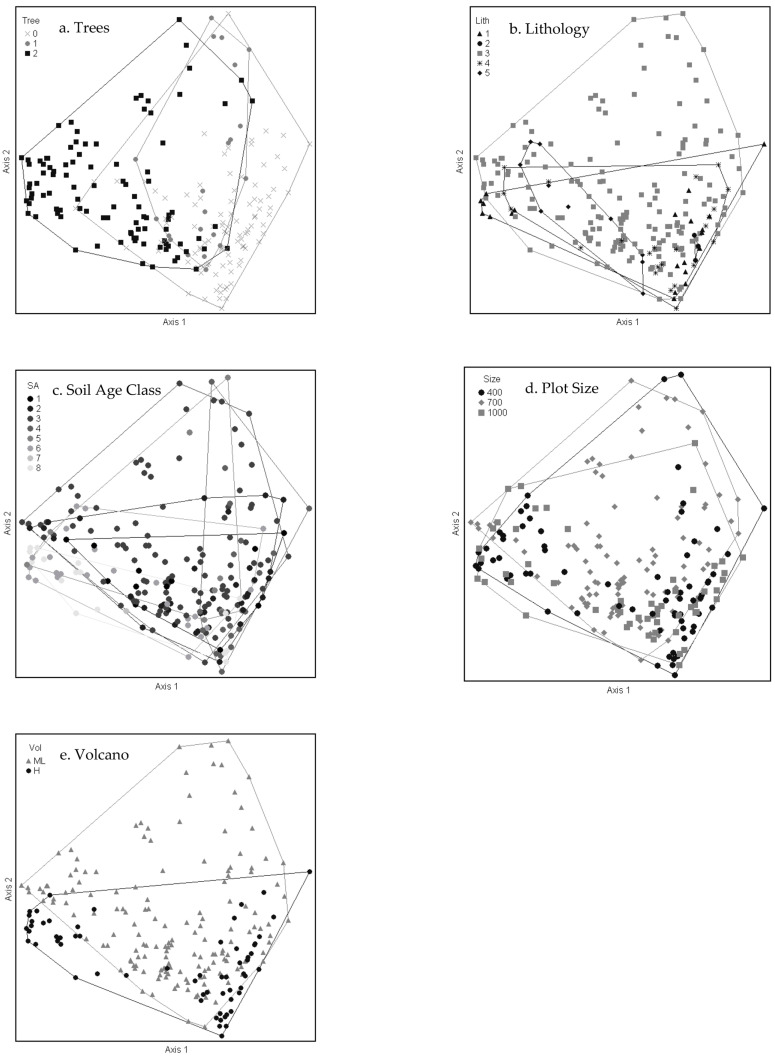
Diagrams of NMS ordination analysis along axes 1 and 2 showing the sample plots grouped by categorical variables: presence of tree life forms > 3 m in height (**a**), 0 = treeless, 1 = <10% tree cover, 2 = ≥10% tree cover); lithology (**b**), 1 = aa, 2 = pahoehoe, 3 = aa + pahoehoe, 4 = sand + cobbles, 5 = tephra fallout + ash; soil age class (**c**), 1 = <750 year, 2 = 750 < 1500 year, 3 = 1500 < 3000 year, 4 = 3000 < 5000 year, 5 = 5000 < 13,000 year, 6 = 13,000 < 30,000 year, 7 = 30,000 < 50,000 year, 8 = >140,000 year; sample plot size (**d**), m^2^; and volcano (**e**), ML = Mauna Loa, H = Haleakalā).

**Figure 6 plants-13-00123-f006:**
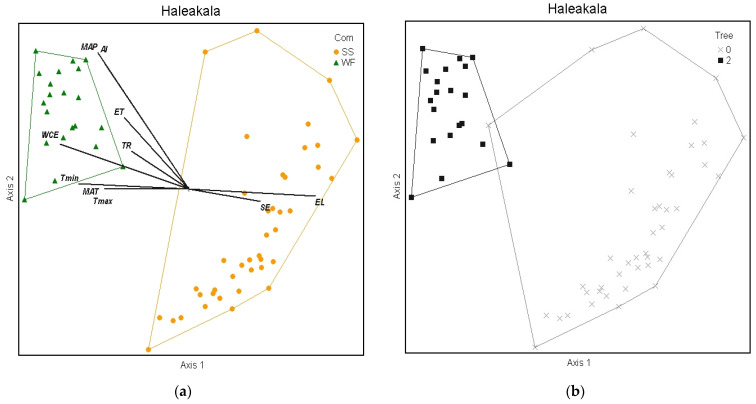
Diagram of NMS ordination analysis along axes 1 and 2 showing the positions of separated cluster groups (SS = subalpine shrubland and WF = wet forest) on the Haleakalā volcano. Significant environmental variable joint plot overlays (see Table 3) vividly display the differences between communities (**a**). Grouping by plots with trees > 3 m closely resembles the community diagram (**b**).

**Figure 7 plants-13-00123-f007:**
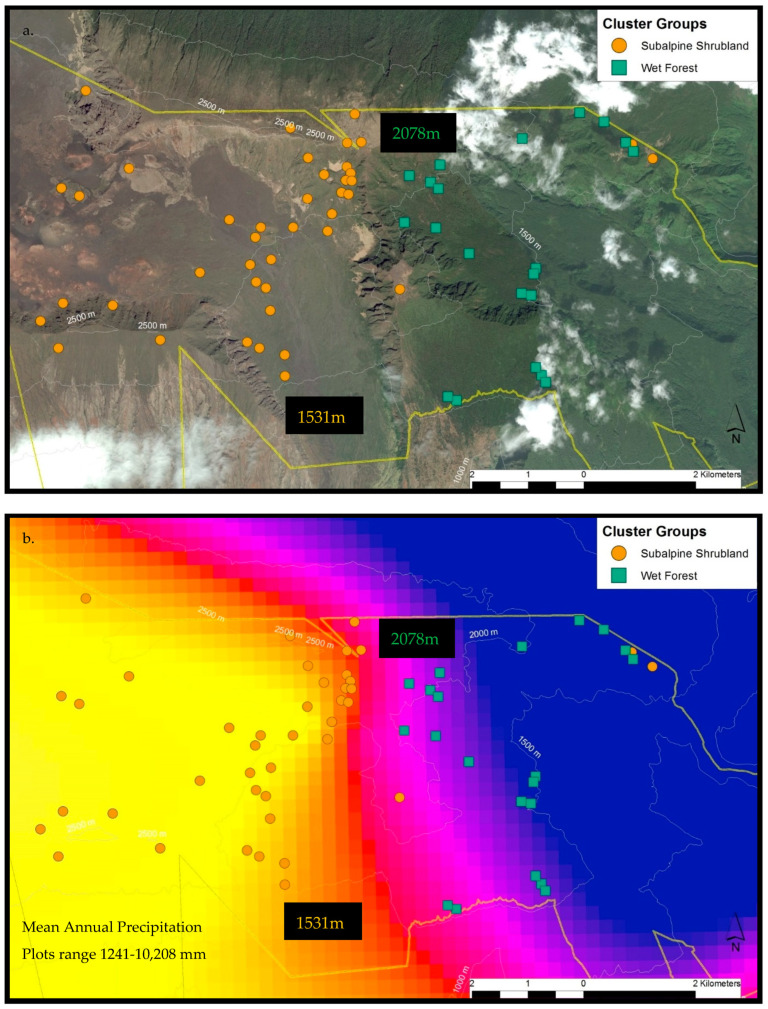
Haleakalā treeline ecotone sample plots by cluster group with the Haleakalā National Park boundary in yellow. On the windward side (northeast aspect), wet forest community plots reach elevations above 2078 m before transitioning into the subalpine shrubland, whereas on the leeward side (southeast aspect), the subalpine community persists below 1531 m (**a**). This site-specific spatial pattern corresponds with modelled mean annual rainfall (**b**), with three exceptionally wet subalpine plots. Two are within rare bog sites (blue-circled) and the other (red-circled) is presumably limited by cold temperatures (**c**).

**Figure 8 plants-13-00123-f008:**
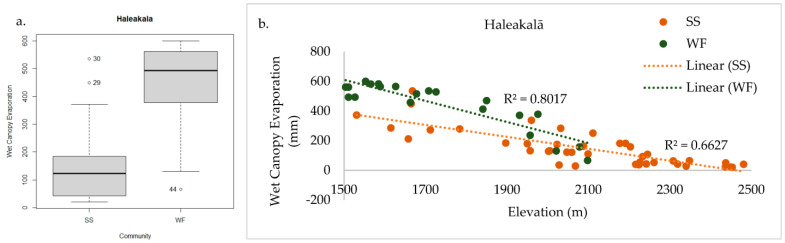
Haleakalā WF had significantly higher wet canopy evaporation than SS (**a**). The threshold where WF transitions to SS is between 200 and 400 mm for wet canopy evaporation, which decreases with elevation (**b**).

**Figure 9 plants-13-00123-f009:**
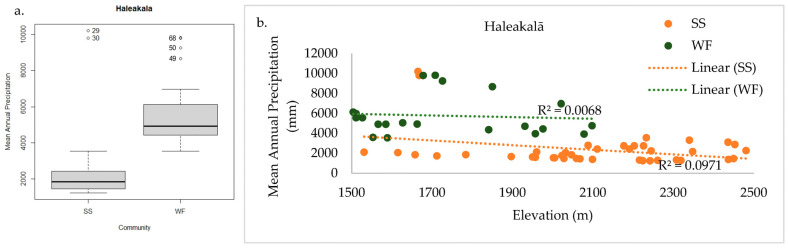
Haleakalā WF had significantly higher mean annual precipitation than SS (**a**). The threshold where WF transitions to SS is approximately 4000 mm mean annual precipitation, independent of elevation (**b**).

**Figure 10 plants-13-00123-f010:**
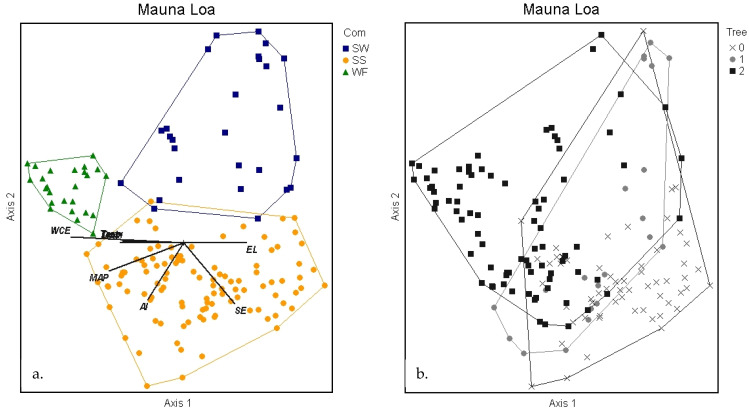
Diagram of NMS ordination analysis along axes 1 and 2 showing the positions of separated cluster groups (SW = subalpine woodland, SS = subalpine shrubland, and WF = wet forest) on the Mauna Loa volcano. Significant environmental variable joint plot overlays (see Table 3) display similar patterns between communities to the full dataset (**a**). Grouping by plots with trees > 3 m (**b**) closely resembles the full dataset plot (Figure 5a) in that trees are not as sensitive an indicator as community composition.

**Figure 11 plants-13-00123-f011:**
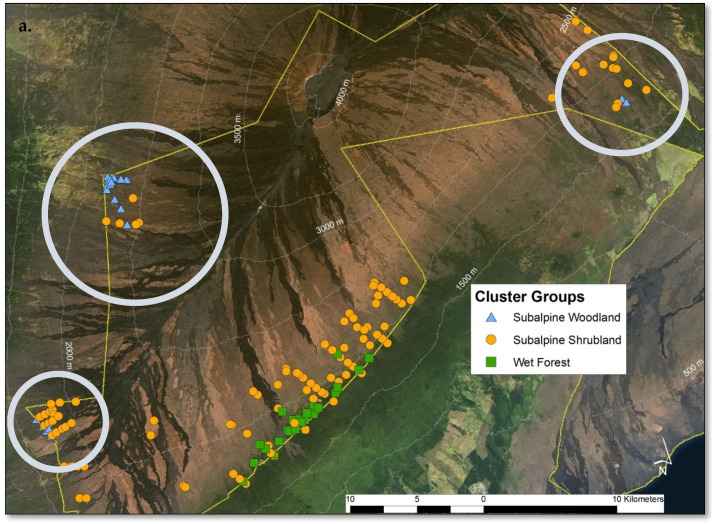
Mauna Loa treeline ecotone sample plots by cluster group with the Hawai’i Volcanoes National Park boundary in yellow. Plots in the subalpine woodland cluster group were only found on Mauna Loa adjacent to subalpine shrublands (**a**). The treeline ecotone between wet forest and subalpine shrubland corresponded with mean annual precipitation, like Haleakalā (**b**). The subalpine woodland was only found in dry sites, and the transition to subalpine shrubland may be correlated with local site temperature (**c**).

**Figure 12 plants-13-00123-f012:**
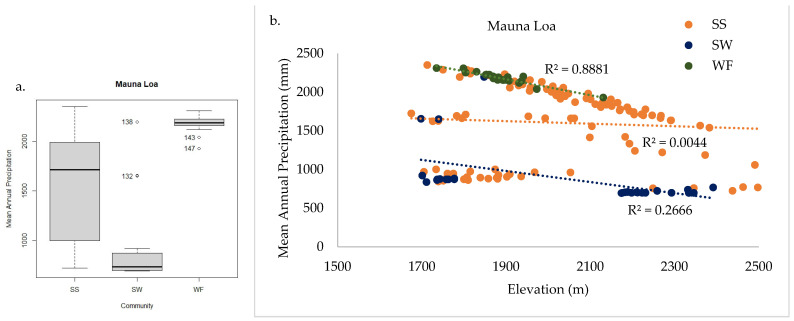
Mauna Loa mean annual precipitation significantly differed among communities (**a**), with high values in WF and low values in SW. The threshold where WF transitions to SS is approximately 2000 mm mean annual precipitation, independent of elevation (**b**). Thresholds for SW mean annual precipitation are less clear because site differences within the community are stronger than patterns between SW and SS communities.

**Figure 13 plants-13-00123-f013:**
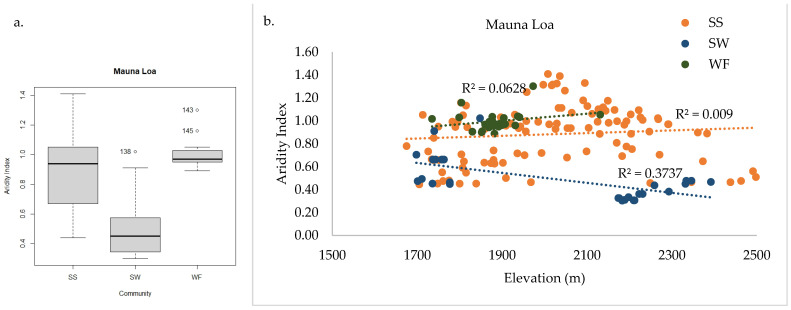
Mauna Loa aridity index values differed among communities (**a**), with significantly lower values in SW. The threshold where WF transitions to SS is, approximately, a 1.0 aridity index, independent of elevation (**b**). Thresholds for SW aridity index are less clear because site differences within the community are stronger than patterns between SW and SS communities.

**Figure 14 plants-13-00123-f014:**
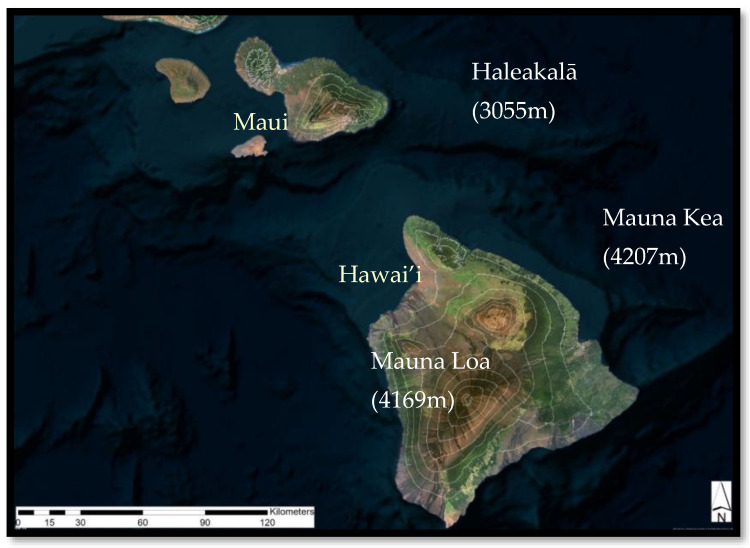
Three Hawaiian volcanic mountains support extremely young, trade wind, inversion-driven treeline ecotones.

**Table 1 plants-13-00123-t001:** Community diversity measures. Alpha (α) diversity is species richness per sample unit; beta (βw) diversity (=(γ/α) − 1) assumes no specific environmental gradient and represents the approximate number of distinct types; and gamma (γ) diversity represents the richness of species across all sample plots in each community. Gamma diversity is further divided by species’ biogeographic origin (BO) (end = endemic, ind = indigenous, non = non-native). Shannon–Wiener (H’) and Simpson’s (D’) diversity indices are rank scales and incorporate species abundance.

Community	N	*α*	*β_w_*	*γ*	*γ* by BO			*H′*	*D′*
					End	Ind	Non		
Subalpine Shrubland	152	13.2 ± 4.7 *	8.7	128	39%	16%	45%	1.524	0.6754
Subalpine Woodland	27	15.4 ± 7.9 *	5.2	96	30%	13%	57%	1.657	0.6916
Wet Forest	46	25.1 ± 10.2 *	4.2	131	67%	20%	13%	1.898	0.7267

* standard deviation.

**Table 2 plants-13-00123-t002:** Indicator species. Significant indicator species of the wet forest, subalpine shrubland, and subalpine woodland cluster groups ranked by indicator value (IV). Life form, biogeographic origin (BO) (end = Hawaiian Islands endemic, ind = indigenous, non = non-native), and Jaccard habitat specialization index (JHS) values are included. JHS values range from specialist (0.5) to generalist (1.0).

Species	Life Form	BO	JHS	IV
**Wet Forest**				
*Cheirodendron trigynum* ssp. *trigynum* (Gaudich) A. Heller	tree	end	0.787	99.5
*Elaphoglossum wawrae* (Luerss.) C. Chr.	fern	end	0.753	91.6
*Dryopteris wallichiana* (Spreng.) Hyl.	fern	ind	0.841	87.6
*Uncinia uncinata* (L. f.) Kük.	sedge	ind	0.746	86.9
*Vaccinium calycinum* Sm.	shrub	end	0.797	82.3
*Myrsine lessertiana* A. DC.	tree	end	0.741	76.4
*Elaphoglossum paleaceum* (Hook. and Grev.) Sledge	fern	ind	0.761	74.7
*Ilex anomala* Hook. and Arnott	tree	ind	0.768	73.7
*Metrosideros polymorpha* Gaudich.	tree	end	0.876	73.5
*Athyrium microphyllum* (Sm.) Alston	fern	end	0.751	65.0
*Rubus hawaiensis* A. Gray	shrub	end	0.765	64.6
*Sadleria pallida* Hook. and Arn.	tree fern	end	0.755	58.5
*Carex alligata* Boott	sedge	end	0.755	53.8
*Cibotium glaucum* (Sm.) Hook. and Arn.	tree fern	end	0.824	53.2
*Diplazium sandwichianum* (C. Presl) Diels	fern	end	0.757	50.0
*Thelypteris globulifera* (Brack.) C. F. Reed	fern	end	0.610	49.7
**Subalpine Shrubland**				
*Kadua affinis* DC.	tree	end	0.748	41.3
*Astelia menziesiana* Sm.	herb	end	0.728	39.0
*Coprosma foliosa* A. Gray	shrub	end	0.748	38.5
*Melicope clusiifolia* (A. Gray) T. G. Hartley and B. C. Stone	tree	end	0.762	36.8
*Coprosma ochracea* W. R. B. Oliv.	tree	end	0.752	36.2
*Broussaisia arguta* Gaudich.	shrub	end	0.762	34.8
*Dryopteris rubiginosa* (Brack.) Kuntze	fern	end	0.681	28.3
*Lepisorus thunbergianus* (Kaulf.) Ching	fern	ind	0.803	27.9
*Smilax melastomifolia* Sm.	vine	end	0.757	26.1
*Cyclosorus sandwicensis* (Brack.) Copel.	fern	end	0.701	21.7
*Vaccinium reticulatum* Sm.	shrub	end	0.803	84.3
*Deschampsia nubigena* Hillebr.	grass	end	0.755	69.4
*Morelotia gahniiformis* Gaudich.	sedge	end	0.715	65.4
*Leptecophylla tameiameiae* (Cham. and Schltdl.) C. M. Weiller	shrub	ind	0.842	63.2
*Coprosma ernodeoides* A. Gray	shrub	end	0.712	62.4
*Hypochoeris radicata* L.	herb	non	0.792	54.2
*Dodonaea viscosa* Jacq.	shrub	ind	0.808	35.5
*Luzula hawaiiensis* var. *hawaiiensis* Buchenau; Buchenau	rush	end	0.716	33.2
*Pellaea ternifolia* (Cav.) Link	fern	ind	0.732	33.1
*Holcus lanatus* L.	grass	non	0.792	33.0
*Pteridium aquilinum* (L.) Kuhn	fern	ind	0.796	31.0
*Coprosma montana* Hillebr.	shrub	end	0.684	24.8
*Trisetum glomeratum* (Kunth) Trin.	grass	end	0.775	21.7
*Eragrostis brownei* (Kunth) Nees ex Steud.	grass	non	0.741	20.4
**Subalpine Woodland**				
*Ehrharta stipoides* Labill.	grass	non	0.860	98.2
*Myoporum sandwicense* A. Gray	shrub/ tree	ind	0.828	48.1
*Cirsium vulgare* (Savi) Ten.	herb	non	0.823	46.1
*Sophora chrysophylla* (Salisb.) Seem.	shrub/ tree	end	0.807	42.3
*Geranium homeanum* Turcz.	herb	non	0.768	40.2
*Verbascum thapsus* L.	herb	non	n/a	39.2
*Rumex acetosello* L.	herb	non	0.790	37.5
*Cenchrus clandestinus* Hochst. Ex Chiov.	grass	non	0.853	37.0
*Pseudognaphalium sandwicensium* (Gaudich.) A. Anderb.	herb	end	0.829	34.7
*Senecio sylvaticus* L.	herb	non	0.735	33.2
*Oxalis corniculata* L.	herb	non	0.895	32.7
*Acacia koa* A. Gray	tree	end	0.854	30.7
*Veronica plebeia* R. Br.	herb	non	n/a	27.7
*Cardamine flexuosa* With.	herb	non	n/a	25.9
*Veronica serpyllifolia* L.	herb	non	0.776	24.7

**Table 3 plants-13-00123-t003:** Environmental correlations. Pearson and Kendall correlations (*r*) of environmental variables with the NMS ordination axes presented in Figure 2. The squared values (*r*^2^) of the correlation coefficients express the proportion of variation in position on an ordination axis that is explained by the variable in question. Strong correlations are highlighted with bold font.

	NMS—Axes					
Environmental Variables	1		2		3	
	*r*	*r* ^2^	*r*	*r* ^2^	*r*	*r* ^2^
Wet canopy evaporation	−0.673	**0.453**	0.022	0.000	0.092	0.008
Mean annual precipitation	−0.525	**0.276**	−0.159	0.025	−0.056	0.003
Elevation	0.504	**0.254**	−0.041	0.002	−0.257	0.066
Minimum temperature	−0.490	**0.240**	0.096	0.009	0.275	0.076
Soil evaporation	0.474	**0.225**	−0.328	0.108	−0.080	0.006
Aridity index	−0.473	**0.224**	−0.186	0.034	−0.037	0.001
Transpiration	−0.441	**0.194**	−0.114	0.013	−0.024	0.001
Mean annual temperature	−0.433	**0.187**	0.103	0.011	0.273	0.075
Maximum temperature	−0.391	0.153	0.183	0.033	0.306	0.094
Potential evapotranspiration	−0.392	0.154	0.003	0.000	−0.140	0.020
Evapotranspiration	−0.329	0.108	−0.416	**0.173**	−0.003	0.000

## Data Availability

Vegetation data are compiled from long-term monitoring plots (DataStore—Pacific Island Network Focal Terrestrial Plant Communities Monitoring Dataset (https://www.nps.gov/index.htm (accessed on 14 October 2023)) and vegetation mapping plots from Hawaii Volcanoes NP (https://irma.nps.gov/DataStore/Reference/Profile/2230292 (accessed on 14 October 2023)) and Haleakalā NP (https://irma.nps.gov/DataStore/Reference/Profile/2230273 (accessed on 14 October 2023)). Environmental variables for each sample site are provided in the Appendix A. Additional data sharing is available upon request.

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
