# Peer review of "Hawaiian Treeline Ecotones: Implications for Plant Community Conservation under Climate Change"

_plants, 2023, doi:10.3390/plants13010123_

Round 1
Reviewer 1 Report
Comments and Suggestions for Authors
After an initial review of the paper, I have identified sections that are absent in the submitted manuscript, possibly due to an oversight during submission. There are no 'Study Area' and 'Methods' sections, which explain my low rate of the 'Quality of Presentation' part. I believe that inclusion of these parts is imperative for a correct evaluation of the work. These sections should address fundamental questions such as where and how the research was conducted, data acquisition procedures, details on the number and size of plots used, a brief description of the geographical and climatic context, etc. All of these data are essential for an understanding of the subsequent scientific interpretations.
Furthermore, I find the paper's subject matter to be of interest to the Plants audience, and I feel that data interpretation align well with the research's objectives and it is clear and appropriate. So it has the potential to become a good work after addressing these missing sections.
However, again, to conduct a proper evaluation of the manuscript, I believe that a complete article is needed.
Author Response
The Plants journal template includes the Material and Methods section after the Discussion section. We suspect Reviewer #1 did not notice this formatting style and stopped reviewing when there was no methods or site description between the introduction and results sections? We look forward to more comments from this reviewer - some suggestions on how to make the discussion section more interesting as opposed to simply site details would be great. Thanks.
Reviewer 2 Report
Comments and Suggestions for Authors
The authors present an interesting study on the sensitivity of species in tropical alpine ecotones in Hawaii to climate variability. It highlights the need to understand the diverse characteristics of lineage ecotones across different volcanoes and climatic conditions. The study, based on data from Haleakalā and Mauna Loa, highlights the importance of tree line indicator species in wet forest, subalpine forest, and subalpine scrub, suggesting that their frequency or abundance may better predict climate change than the presence or absence of tree life forms. The study supports the idea that changes in available moisture, rather than temperature, will likely shape the future of ecotone communities in the Hawaiian treeline.
The introduction is well written and introduces the reader to the subject matter very well. Also, both previous studies and objectives are well outlined.
However, the description of the Hawaiian Islands should be moved to the Material and Methods section. Also, the map should be in the same section.
Results:
135 What is MRPP analysis or NMS ordination? The reader won't understand until they get to the end (in the Material and Methods section).
It is stated that based on species composition the three groups have been named but at least a bibliographic reference is needed.
How were the indicator species chosen? Why are these species considered indicator species?
Table 2 shows unnecessary details: family and common name. It is sufficient to mention how many species belong to a family.
Discussion
The discussions are mainly based on the description of the studied sites and not on the results. However, these are overrated.
Material and Methods
It is not clear how the data were obtained. They also do not appear in the annex, so the results are impossible to reproduce.
I would recommend that the authors consider Raunkiaer's classification of life forms.
Also, the biogeographical origin is presented in a simplistic form.
Conclusions: Besides the introduction it is the best written section.
In conclusion, the idea of this study is very inspiring. Also, there are several well written sections and the graphs are suggestive. The data can be improved and this study can be a reference in the field.
Author Response
Thank you for your useful suggestions. We incorporated your suggestions into the revised version including the following:
Moved Hawaiian section and map figure from Introduction to the Materials and Methods section.
lines 94 and 102 – added Hawaiian since the removal of the previous paragraph.
Updated all figures and references after moving the Hawaiian section and figure.
Defined and added citations for results (MRPP, NMS ordination, Indicator Species) since the methods descriptions are at the end of the manuscript.
Table 2 – removed Family and common names. Added “Hawaiian Islands” before endemic in caption as none of the endemic species listed as indicators were single-island endemics. This is addressing the reference to simplistic biogeographic origin. Also added reference in methods that life form and biogeographic origin are from the Hawaiian flora. As a note, Raunkiaer's classification of life forms is not recommended by some authors for tropical environments (e.g., J. Ewel).
Added Supplemental Materials including (S1) method for combining different datasets and (2) environmental variable values for all plots.
Vegetation data used in the analyses are available to the public and the links are included in the Data Availability Statement (lines 650-655).
We would appreciate any more specific sections for how to make the discussion more interesting and less of an extensive site description.
Round 2
Reviewer 2 Report
Comments and Suggestions for Authors
It is obvious that the ms has been improved. I have noted that the authors have responded to the comments provided. I have no further comments.
Author Response
Thank you, Reviewer 2, for your comments and acknowledgement of our improvements to the manuscript.